# Solid-Phase Synthesis of an Insect Pyrokinin Analog Incorporating an Imidazoline Ring as Isosteric Replacement of a *trans* Peptide Bond

**DOI:** 10.3390/molecules26113271

**Published:** 2021-05-28

**Authors:** Krzysztof Kaczmarek, Barbara Pacholczyk-Sienicka, Łukasz Albrecht, Janusz Zabrocki, Ronald J. Nachman

**Affiliations:** 1Insect Control and Cotton Disease Research Unit, ARS, U.S. Department of Agriculture, 2881 F-B Road, College Station, TX 77845, USA; janusz.zabrocki@p.lodz.pl; 2Institute of Organic Chemistry, Lodz University of Technology, 90-924 Łódź, Poland; barbara.pacholczyk@p.lodz.pl (B.P.-S.); lukasz.albrecht@p.lodz.pl (Ł.A.)

**Keywords:** insect neuropeptides, pyrokinins, *trans* peptide bond, imidazoline ring, SPOS

## Abstract

A facile solid-phase synthetic method for incorporating the imidazoline ring motif, a surrogate for a *trans* peptide bond, into bioactive peptides is reported. The example described is the synthesis of an imidazoline peptidomimetic analog of an insect pyrokinin neuropeptide via a cyclization reaction of an iminium salt generated from the preceding amino acid and 2,4-diaminopropanoic acid (Dap).

## 1. Introduction

Imidazoline moiety has been previously introduced by Jones and colleagues as a peptide bond isostere (with an amidine as an amide bond replacement) [1,2]. Nachman et al. proposed that the imidazoline moiety can specifically function as a mimic or surrogate of the *trans* peptide bond, including a *trans* Pro, locking a *trans* orientation within the constrained five-membered imidazoline ring (Figure 1) [3,4,5].

However, whereas the molecular modeling suggests that the imidazoline moiety can function as a mimic of a *trans* Pro, it is also clear that it is not an exact mimic [3,4,5]. Therefore, analogs containing imidazoline moiety provide an opportunity for either a selective agonist interaction with closely related receptors of a superfamily or as an antagonist, as some receptors may display more tolerance to small deviations from the *trans* peptide bond and/or *trans* Pro structure of natural peptides than others.

The pyrokinin (PK) family of peptides plays a multifunctional role in the physiology of insects. In 1986, the first member of the family, leucopyrokinin (LPK), was isolated from the cockroach *Leucophaea maderae* [6] with over 30 members of this peptide class identified thereafter. All family members share the common *C*-terminal pentapeptide FXPRL-amide (X = S, T, G or V) and include subfamilies such as PKs, myotropins (MTs), PBAN, diapause hormone (DH), melanization and reddish coloration hormone (MRCH), pheromonotropin (PT), and pheromonotropic *β* and *γ* peptides derived from the cDNA of moths [7,8,9,10,11,12]. The PK family has been shown to stimulate sex pheromone biosynthesis in moths [7,11,12] and mediate critical functions associated with feeding (gut contractions) [10], development (egg diapause, pupal diapause, and pupariation) [12,13,14,15,16,17], and defense (melanin biosynthesis) [18] in a variety of insects.

## 2. Results and Discussion

Incorporation of the imidazoline moiety into a PK/PBAN C-terminal hexapeptide sequence led to an analog, labeled PPK-Jo (Ac-YF[Jo]RLa) (Figure 2) [3,5]. PPK-Jo demonstrated strong activity in an in vivo *S. littoralis* melanotropic assay, matching the efficacy of PBAN and leucopyrokinin (LPK). Unlike the parent PK/PBAN hexapeptide YFTPRLa, PPK-Jo is a pure melanotropic agonist in the *S. littoralis* assay. PPK-Jo failed to elicit significant agonist (or antagonist) activity in three other related PK bioassays, i.e., an in vivo *H. peltigera* pheromonotropic assay, an in vivo *N. bullata* pupariation assay, and in an in vitro cockroach *L. maderae* hindgut myotropic assay [3,5].

Another member of the PK neuropeptide family, the diapause hormone (DH), has been shown to terminate the protective state of diapause in pupae of heliothine moths [15]. The imidazoline motif was incorporated into the C-terminal heptapeptide active core region (LWFGPRLa) of DH [4,5], transforming a DH agonist into an antagonist capable of blocking the diapause termination activity of the native hormone in the corn earworm moth *H. zea*. The analog provides a lead for the development of DH analogs that can control insect pests via disruption of diapause behavior.

In this paper, we describe a solid-phase peptide synthetic (SPPS) route to incorporation of the imidazoline ring into the PK analog PPK-Jo (Ac-YF[Jo]RLa) (Figure 2). Solid-phase synthesis on a polymer support offers advantages in that laborious purification at intermediate steps is eliminated and replaced by simple washing and filtration. In addition, the entire synthesis can be carried out in a single vessel, minimizing the loss of material during transfer from one vessel to another that would occur in solution-phase synthesis. We decided to obtain the imidazoline motif through a cyclization reaction of an iminium salt generated from the preceding amino acid amide and 2,4-diaminopropanoic acid (1,3-diaminopropionic acid, Dap). Considering available solution-phase syntheses leading to such rings that have been previously described [1,2] which are very laborious, we decided to elaborate on the general procedure to allow the synthesis of such units in SPPS. Such a method would be very convenient for incorporation of the imidazoline unit into a wide range of peptide analogs.

We started the synthesis with consecutive couplings of Fmoc-Leu-OH and Fmoc-Arg(Pbf)-OH to Fmoc-Rink-amide resin by means of HBTU/HOBt/DIEA as a reagent in NMP as a solvent, using a 0.25 mmol scale protocol (ABI) on an ABI 440 Peptide Synthesizer. The protocol for either manual or automated machines usually consists of the following repeated operations for every single amino acid residue to be added to the growing peptide chain. These steps are as follows: 1. deprotection of the amino group of the amino acid, followed by a washing step; 2. coupling (acylation) reaction of the next protected amino acid, followed by a washing step; 3. capping the unreacted, unacylated amino group, followed by a washing step.

After obtaining the dipeptide H-Arg(Pbf)-Leu- attached to Rink amide resin, we attempted to perform the coupling of Fmoc-Dap(Fmoc)-OH under the same conditions. Unfortunately, due presumably to the low solubility of the bis-Fmoc-1,3- diaminopropanecarboxylic acid derivative in NMP, the coupling step performed on the ABI synthesizer was not sufficiently efficient, and this step had to be performed manually in a polypropylene syringe. After simultaneous removal of both fluorenylmethyleno-xycarbonyl (Fmoc) groups (20% piperidine/NMP, then washing with NMP (3×) and MeOH (3×) and drying), both free amino groups were then allowed to react with a solution of iminium salt **2** derived from Fmoc-Ala-NH_2_ **1** and triethyloxonium hexafluorophosphate or tetrafluoroborate (Figure 3). This reaction was also performed manually in the syringe. Fmoc-Ala-NH_2_ 1 was synthesized from commercially available substrates and reagents, either from Fmoc-Ala-OSu by dissolving it in MeOH containing 1.1 eq. ammonia or from H-Ala-NH_2_ and Fmoc-OSu overnight in dioxane on a scale up to 20 mmol. After overnight stirring at rt followed by pouring into five volumes of water, product **1** quantitatively precipitated. The precipitate was collected by simple filtration and was extensively washed with water to remove the side product N-hydroxysuccinimide. Crude **1** was dried over phosphorus pentoxide in a desiccator to yield a white powder (MH^+^ 311, Kratos) with a purity of over 99% (HPLC) and was used for generation of the iminium salt as is.

Surprisingly, the imidazoline ring formation performed in DCM went well in contrast to NMP, which did not give the expected product. Reaction was carried out overnight manually before the removal of the Fmoc from the N-terminal nitrogen; the secondary nitrogen atom of the imidazoline ring was orthogonally protected to avoid an undesired acylation during the next coupling reaction. We checked the utility of two N-protecting groups—tert-butyloxycarbonyl (Boc) and trityl (Trt). They were introduced through a reaction with tert-butyl pyrocarbonate or trityl chloride in the presence of a tertiary amine, respectively. We found that during the cleavage of the pseudopeptide from the resin and final side chain deprotection, Boc protection gave products with higher yield and better purity.

Our synthetic route should not pose a larger racemization issue during SPPS than in other manual and automated SPPS syntheses, because we used HBTU as coupling reagent and HOBt as additive, which should suppress racemization. Moreover, the amino acids we used were protected with a urethane type group—fluorenylmethylenoxycarbonyl (Fmoc). Our iminoether should be less prone to epimerization due to urethane-type protecting group. There is no reaction that involves the breaking of any bonds to a chiral center during cyclization to the imidazoline ring, and the alpha carbon atom of the Alanine residue is not a part of the imidazoline ring. Therefore, danger of epimerization of this chiral center should be diminished or even avoided. In the Supporting Information (SI), we have provided an HPLC chromatogram from the final purification step that shows a very small peak preceding the main product that features the same parent ion in the mass spectrum. We had surmised that this was likely an epimer and that this may indicate that the threat of racemization is minimal.

After the secondary imidazoline nitrogen was protected, the next standard steps for synthesizing the rest of the pseudopeptide sequence can be completed either manually in a syringe or on an automated peptide synthesizer. The cleavage of the final pseudopeptide from the resin with simultaneous deprotection of side chains was first attempted under standard conditions: a cocktail composed of TFA/TIS/water (95/2.5/2.5), 2–4 h, rt, which gave no expected product, either due to reduction of the imidazoline double bond by silane or due to hydrolysis (presence of water in a cleavage cocktail). We found that a cocktail composed of TFA/DCM (1:1), 2–3 h, rt, used generally for the removal of acid labile groups such as Boc, t-Bu, and Trt in solution-phase synthesis worked well for the cleavage of this pseudopeptide–resin complex. The final pseudopeptide product was purified on a Waters C_18_ Sep Pak cartridge and Delta-Pak C_18_ reverse-phase columns. After lyophilization of combined fractions, a white, fluffy solid was obtained. The identity of the analog during the purification steps was confirmed via MALDI-MS on a Kratos Kompact Probe MALDI-MS machine (Kratos Analytical, Ltd., Manchester, UK) with the presence of the molecular ion (778.11 [MH^+^]). The structure was additionally confirmed after purification by high-resolution mass spectrometry (HRMS) and NMR (see Appendix A). Calc. for C_38_H_56_N_11_O_7_ = 778.4264 [MH^+^] found 778.4254 [MH^+^] and 379.7226 [MH_2_^2+^] for single and doubly charged ions, respectively. The presence of the imidazoline ring was confirmed based on 2D NMR spectroscopy. On the HMBC spectrum, we observed a correlation peak between the methine proton (CH, 4.82 ppm) and the quaternary carbon atom of the imidazoline ring (C=N, 146.50 ppm). Moreover, this carbon also has correlations with CH_2_ (3.72, 4.10 ppm), CH_3_ (1.44 ppm), and CH (4.08 ppm) groups (Figure 4).

Yield of the crude peptide attached to the polymer support, calculated from weight gain, was approximately 90–95% in several trials on the scale from 0.05 mmol to 0.125 mmol. Yield: 9–12% total of repurified product (HPLC) depending on the scale of synthesis. For the scale of 0.05 mmol, the yield was 10% (5.12 micromoles).

## 3. Materials and Methods

All solvents purchased were HPLC or anhydrous p.a. grade (SIGMA-Aldrich, Milwaukee, WI, USA and Milipore, Danvers, MA, USA) and were used without purification. Fmoc-Ala-OSu, Fmoc-OSu, H-Ala-NH_2_, Rink resin, HBTU, HOBt, DIEA, and Fmoc-Dap(Fmoc)-OH were purchased from ChemImpex, Inc. (Wood Dale, IL, USA) or IRIS Biotech (Marktredwitz, Germany). All other organic reagents were supplied by Aldrich (Milwaukee, WI, USA).

General automated SPPS (on ABI 433A Peptide Synthesizer; Applied Biosystems Inc., Waltham, MA, USA):

Rink resin (0.25 mmol) was initially swollen in NMP with vortexing for 10 min at rt. (i) Deprotection (Fmoc group removal): 5 mL of 20% piperidine in NMP 3 × 2.5 min while vortexing, rt. Completeness of deprotection was checked comparing the difference in the value of conductivity between the last deprotection step with the preceding one with a conductivity meter, and if this value was higher than the standard value, which was set up in programming, the machine was repeating deprotections (up to three more for difficult deprotections). (ii) Washing: 6 × 5 mL NMP. (iii) Coupling: Fmoc-AA-OH (1 mmol, 4 eq.) in the cartridge, to which HBTU/HOBt (3.6 eq., 0.45 M in DMF) and DIEA (7.2 eq., 2.0 M in NMP) were added; the solution in the cartridge was agitated with a gentle stream of nitrogen for 10 min, after which it was transferred to the washed resin. The resulting suspension was vortexed for 45 min at rt. The reaction vessel was then drained, and the resin was thoroughly washed with 5 mL NMP three times while vortexing. (iv) Capping: 5 ml of NMP solution containing acetic anhydride/DIEA/HOBt (19 mL/9.5 mL/800mg diluted with NPM to 400 mL) was added to the resin, and the reaction vessel was vortexed for 15 min and drained, and the resin was thoroughly washed with 5 mL DCM six times.

General Peptide Cleavage Method: The resin was soaked with DCM and cleaved using 50% TFA/DCM cleavage cocktail at rt for 2 h. A cocktail, which is mostly used for this purpose, TFA/TIS/water, 95:2.5:2.5, was found to be not compatible with imidazoline moiety. The resin was then filtered from the cleavage cocktail solution, and the solution was concentrated by evaporation under vacuum. The residual liquid was poured into diethyl ether (at least 10 times the volume of the residual solution) to precipitate a crude product. The crude peptide was sedimented by gentle centrifugation, and then, the ether phase was decanted. Suspending of the crude peptide in ether and decantation was repeated twice. The crude peptide PPK-Jo was dissolved in 20% acetonitrile in water and purified by reverse-phase preparative HPLC using method A (below). Fractions containing peptides of expected molecular weight were dried, dissolved in 80% acetonitrile/water, and further purified by normal-phase preparative HPLC using method B.

All NMR spectra of the obtained compound were acquired using a Bruker Avance II Plus 16.4 T spectrometer (Bruker BioSpin, Germany). All experiments were performed at 300 K. The operating frequencies were 700 and 175 MHz for ^1^H and ^13^C experiments. The instrument was equipped with a 5 mm Z-gradient broadband decoupling inverse probe. The precise ^1^H and ^13^C chemical shifts assignments were performed by 2D NMR experiments. All chemical shifts were referenced to the DMSO signal at 2.50 ppm. 2D NMR spectra were processed with TopSpin 2.1 (Bruker).

HRMS measurements were performed using a Synapt G2-Si mass spectrometer (Waters Corporation, Milford, MA, USA) equipped with an ESI source and quadrupole time-of-flight mass analyzer. The mass spectrometer was operated in the positive ion detection mode. To ensure accurate mass measurements, data were collected in centroid mode, and the mass was corrected during acquisition using leucine enkephalin solution as an external reference (Lock-SprayTM), which generated reference ion at m/z 556.2771 Da [M+H]^+^ in positive ESI mode. The results of the measurements were processed using the MassLynx 4.1 software (Waters) incorporated with the instrument.

The identity of the analog during purification steps was confirmed via MALDI-MS on a Kratos Kompact Probe MALDI-MS machine (Kratos Analytical Ltd., Manchester, UK).

### 3.1. Experiment

#### 3.1.1. Fmoc-Ala-NH_2_ **1**

**Method A**. To the solution of Fmoc-Ala-OSu (4.084g, 10 mmol, IRIS Biotech, Germany) in methanol (50 mL), dropwise solution of 7N ammonia in methanol was added at rt (1.5 mL, 10.5 mmol, 1.05 eq., Aldrich). The reaction solution was then stirred overnight, and it was poured into 250 mL of cold water while stirring. After sitting overnight in refrigerator, a white solid was filtered, washed thoroughly with water, and dried in a desiccator overnight over P_2_O_5_. Yield 3.073 g (99.0%). MW calc. for C_18_H_18_N_2_O_3_ 310.353 found MH^+^ 311.5 (MALDI-MS, KRATOS). RP-HPLC purity on C_18_ column in gradient 40-80% B (B 0.1% TFA/80% acetonitrile/20% water) was over 99%. This product was used in the next step without further purification. CAS 136497-80-8.

This reaction was repeated a few times and scaled up to 50 mmol without changing the quality of received product.

**Method B**. To the solution of Fmoc-OSu (6.746 g, 20 mmol, ChemImpex) in 100 mL of dioxane, H-Ala-NH_2_ (1.762 g, 20 mmol, IRIS Biotech, Germany)) was added and stirred overnight. RP-HPLC showed disappearance of Fmoc-OSu. The reaction solution was poured slowly to 500 mL 0.01 M KHSO_4_ in water to remove traces of residual substrate. The precipitated white solid was filtered and washed thoroughly with water and dried in a desiccator overnight over P_2_O_5_. Yield 6.120 g (98.6%). MW calc. for C_18_H_18_N_2_O_3_ 310.353 found MH^+^ 311.4 (MALDI-MS, KRATOS). RP-HPLC purity on C_18_ column in gradient 40-80% B (B 0.1% TFA/80% acetonitrile/20% water) was higher than 99%. This product was used in the next step without further purification.

#### 3.1.2. Iminoether **2**

Fmoc-Ala-NH_2_ **1** (1.55 g, 5 mmol) was added to a cooled ice-bath solution of triethyloxonium tetrafluoroborate (1.045 g, 5.5 mmol, 1.1 eq., Aldrich) in dry DCM. The resulting suspension was stirred at this temperature for 2 h, after which the suspension disappeared. The reaction mixture was washed once with 1M KHCO_3_ to remove excess reagent, and the DCM layer was dried over anhydrous MgSO_4_ overnight. After filtration, the resulting solution was concentrated on a rotary evaporator under vacuum, quantitatively giving almost colorless oil, which solidified after overnight refrigeration. This product was used in the next step without further purification. During prolonged storage, even as a solid in the refrigerator, the compound slowly decomposes, turning its color to yellow and then orange, so it should be kept in the freezer.

#### 3.1.3. Cyclization Reaction to Imidazoline Ring on Solid Support

To 0.1 mmol H-Dap-Arg(Pmc)-Leu-Rink resin obtained on the ABI Peptide synthesizer, transferred to an 8 mL propylene syringe equipped with 20 um frit, and swollen in dry DCM, a solution of iminoether **2** (0.3 mmol, 3 eq.) in DCM was added. After addition of DIEA (0.05 mmol, 0.5 eq.), the resulting suspension was shaken overnight at rt; filtered; and then washed 3 times with DCM, 3 times with MeOH, and 3 times with DCM. Protection of imidazoline nitrogen was also performed in the syringe, adding a solution of di-*tert*-butyl pyrocarbonate (0.2 mmol, 2 eq., ChemImpex) in DCM to the resin. After being shaken for 3 h at rt, the resin was filtered through frit and washed consecutively with DCM (3×), MeOH (3×) and dried. Then, the peptide–resin was placed again in the reaction vessel of the ABI Peptide Synthesizer, and automated synthesis was continued. After the synthesis, the resin was transferred back to the polypropylene syringe, washed with MeOH (3×), and dried. The yield of the crude peptide attached to the polymer support, calculated from weight gain, was in the range of 90–95% in several trials on the scale from 0.05 to 0.125 mmol.

#### 3.1.4. Purification and Amino Acid Analysis

Method A. A Waters C_18_ Sep Pak cartridge and a Delta-Pak C_18_ reverse-phase column (8 × 100 mm, 15 μm particle size, 100 A pore size) on a Waters 510 HPLC controlled with a Millennium 2010 chromatography manager system (Waters, Milford, MA) with detection at 214 nm at ambient temperature. Solvent A = 0.1% aqueous trifluoroacetic acid (TFA); Solvent B = 80% aqueous acetonitrile containing 0.1% TFA. Conditions: Initial solvent consisting of 20% B was followed by the Waters linear program to 100% B over 40 min; flow rate, 2 mL/min. Delta-Pak C-18 retention time: t_R_ = 4.5 min.

Method B. A Waters Protein Pak I 125 column (7.8 × 300 mm). Conditions: Isocratic using 80% aqueous acetonitrile containing no TFA; flow rate, 2 mL/min. Waters Protein Pak retention time: 6.0 min.

The first purification step was performed on a C_18_ Waters cartridge (Method A). The identity of the product during the first purification steps was confirmed via MALDI-MS on a Kratos Kompact Probe MALDI-MS machine (Kratos Analytical, Ltd., Manchester, UK) with the presence of the molecular ion (778.11 [MH^+^]). The fractions containing a peak of correct MS value were combined, freeze-dried, and subjected to the final purification step (Method B).

In Appendix A, HPLC traces of the partially purified product from freeze-dried fractions dissolved and injected onto the Waters Protein Pak I 125 column can be found. This was the second step of purification (Method B) in an isocratic solvent system of 80% acetonitrile/20% water (normal phase). Fractions were collected by cutting the upper half of the main peak in each injection at the marked position (arrows).

The tiny peak forming in front of the main peak was also collected, showing the same mass, but it was discarded as a possible epimer. After lyophilization of the combined fractions, a white fluffy solid was obtained. The fractions containing a peptide analog were freeze dried again for biological activity studies, NMR, and HRMS.

Amino acid analysis was used to quantify the peptides and to confirm their identity, leading to the following analysis: F[1.0], L[1.1], R[1.2], Y[1.2].

#### 3.1.5. HRMS Measurements

HRMS measurements from the repurified sample were performed using a Synapt G2-Si mass spectrometer (Waters) equipped with an ESI source and quadrupole time-of-flight mass analyzer. The mass spectrometer was operated in the positive ion detection mode. To ensure accurate mass measurements, data were collected in centroid mode, and mass was corrected during acquisition using leucine enkephalin solution as an external reference (Lock-SprayTM), which generated a reference ion at m/z 556.2771 Da ([M+H]^+^) in positive ESI mode. The results of the measurements were processed using the MassLynx 4.1 software (Waters) incorporated with the instrument. Calc. for C_38_H_56_N_11_O_7_ = 778.4264 [MH^+^] found 778.4254 [MH^+^] and 379.7226 [MH_2_^2+^] for single- and doubly charged ions, respectively (Appendix A).

#### 3.1.6. NMR Experiments

All NMR spectra of the obtained compound were acquired using a Bruker Avance II Plus 16.4 T spectrometer (Bruker BioSpin, Germany). All experiments were performed at 300 K. The operating frequencies were 700 and 175 MHz for ^1^H and ^13^C experiments, respectively. The instrument was equipped with a 5 mm Z-gradient broadband decoupling inverse probe. The precise ^1^H and ^13^C chemical shifts assignments were performed by 2D NMR experiments. All chemical shifts were referenced to the DMSO signal at 2.50 ppm. Two-dimensional NMR spectra were processed with TopSpin 2.1 (Bruker). Due to the low concentration of the compound, it was not possible to assign all of the chemical shifts of the quaternary carbons. The presence of the imidazoline ring was confirmed on the basis of 2D NMR spectroscopy. Homonuclear correlated spectra (COSY) were acquired using a standard pulse sequence (cosygpqf). Spectra were recorded with the acquisition of 32 transients for each of the 512 increments with 2K data points. The spectral width for ^1^H was 8417 Hz. Total correlated spectra (TOCSY) were acquired using a standard pulse sequence (mlevph). Spectra were recorded with the acquisition of 48 transients for each of the 512 increments with 2K data points. The spectral width for ^1^H was 8417 Hz. Two-dimensional ^1^H-^13^C HSQC (Appendix A) spectra were recorded using the Bruker pulse sequence hsqcetgpsi2. The spectral widths for ^1^H and ^13^C were 8417 and 31706 Hz sampled with 4096 and 256 complex points, respectively. The number of scans was 64. Two-dimensional ^1^H-^13^C HMBC spectra (Appendix A) were recorded using the Bruker pulse sequence hmbcgplpndqf. The spectral widths for ^1^H and ^13^C were 9090 and 39062 Hz sampled with 4096 and 256 complex points, respectively. The number of scans was 640. All spectra were recorded in two solvents, D_2_O and DMSO.

On the HMBC spectrum (Appendix A), we observed a correlation peak between the methine proton (CH, 4.82 ppm) and the quaternary carbon atom of the imidazoline ring (C=N, 146.50 ppm). Moreover, this carbon also has correlations with CH_2_ (3.72, 4.10 ppm), CH_3_ (1.44 ppm), and CH (4.08 ppm) groups.

^1^H (DMSO-d_6_, 300K) δ, ppm: 0.85 (3H, H^δ^, Leu), 0.89 (3H, H^δ^, Leu), 1.27 (1H, H^γ^, Leu), 1.44 (3H, Imid.), 1.45 (2H, H^γ^, Arg), 1.47 (1H, H^β^, Leu), 1.54 (1H, H^β^, Arg), 1.60 (1H, H^β^, Leu), 1.76 (1H, H^β^, Arg), 1.93 (3H, Ac), 2.83 (1H, H^β^, Phe), 2.92 (1H, H^β^, Phe), 3.03 (1H, H^β^, Tyr), 3.10 (2H, H^δ^, Arg), 3.16 (1H, H^β^, Tyr), 3.42 (2H, NH_2_, Arg), 3.44 (2H, NH_2_, Leu), 3.72 (1H, CH_2ring_, imidazoline), 4.08 (1H, CH, imidazoline), 4.10 (1H, CH_2ring_, imidazoline), 4.23 (1H, H^α^, Tyr), 4.25 (1H, H^α^, Leu), 4.34 (1H, H^α^, Arg), 4.51 (1H, H^α^, Phe), 4.58 (1H, OH, Tyr), 4.82 (1H, CH_ring_, imidazoline), 6.61 (2H, H^γ^, Tyr), 6.95 (2H, H^δ^, Tyr), 7.16 (1H, H^γ^, Phe), 7.23 (1H, H^ε^, Phe), 7.25 (1H, NH, Arg), 7.28 (1H, H^δ^, Phe), 7.48 (1H, NH, imidazoline), 8.00 (1H, NH, Leu), 8.03 (1H, NH, Arg), 8.08 (1H, NH, Tyr), 8.10 (1H, NH, Arg), 8.13 (1H, NH, Phe).

^13^C (DMSO-d_6_, 300K) δ, ppm: 21.50 (CH_3_, Ac), 22.41 (C^δ^, Leu), 22.96 (C^δ^, Leu), 24.19 (C^β^, Arg), 24.71 (CH_3_, imidazoline), 24.79 (C^γ^, Leu), 28.94 (C^γ^, Arg), 39.38 (C^β^, Phe), 39.99 (C^δ^, Arg), 40.41 (C^β^, Leu), 40.91 (C^β^, Tyr), 48.63 (CH, imidazoline), 50.84 (C^α^, Phe), 51.96 (C^α^, Tyr), 52.53 (C^α^, Arg), 53.40 (C^α^, Leu), 53.78 (CH_2ring_, imidazoline), 54.51 (CH_ring_, imidazoline), 114.78 (C^γ^, Tyr), 126.28 (C^ε^, Phe), 127.80 (C^δ^, Phe), 129.13 (C^γ^, Phe), 129.89 (C^δ^, Tyr), 146.50 (C=N, imidazoline), 156.62 (C^ε^, Tyr), 157.97 (C^ε^, Arg), 170.84 (C=O, Ac), 171.07 (C=O, Tyr), 171.67 (C=O, Leu), 171.84 (C=O(NH2)), 172.59 (C=O, Phe), 173.88 (C=O, Arg).

HPLC trace, HRMS, and all NMR spectra can be found in the Supporting Information.

## 4. Conclusions

In conclusion, the described synthetic route provides a solid-phase alternative to the introduction of an imidazoline ring motif as a trans peptide bond surrogate into appropriate peptide sequences. It thus offers a useful tool to peptide chemists seeking to develop peptidomimetic analogs of biologically active peptides that can potentially feature selective and/or antagonist properties.

## Figures and Tables

**Figure 1 molecules-26-03271-f001:**
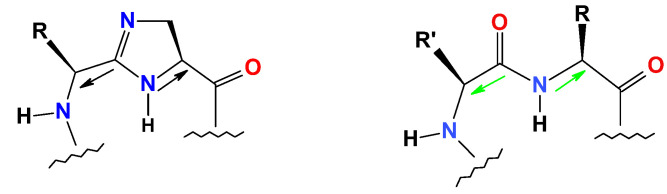
Comparison between *trans* peptide bond structure (**right**) and imidazoline moiety (**left**) as a peptide bond replacement, and which mimics *trans* geometry irreversibly.

**Figure 2 molecules-26-03271-f002:**
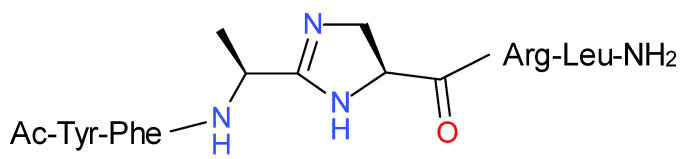
Structure of PPK-Jo, a selective agonist in one of four PK-related bioassays [3,5].

**Figure 3 molecules-26-03271-f003:**
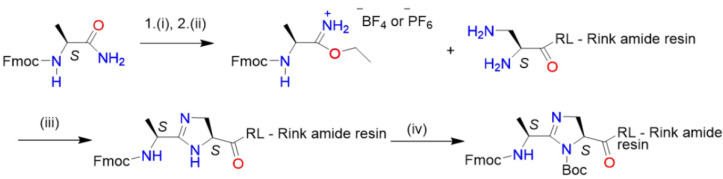
Synthesis of the iminoether **2** derived from Fmoc-L-Ala-NH_2_ **1**, followed by ring formation on solid support: (i) 1.1 eq. triethyloxonium tetrafluoroborate or triethyloxonium hexafluorophosphate in DCM, rt, 2 h; (ii) washing with 1 M KHCO_3_ aq., overnight drying over anhydrous MgSO_4_; (iii) DIEA/DCM, rt, overnight; (iv) Boc_2_O/DIEA, rt, 3 h.

**Figure 4 molecules-26-03271-f004:**
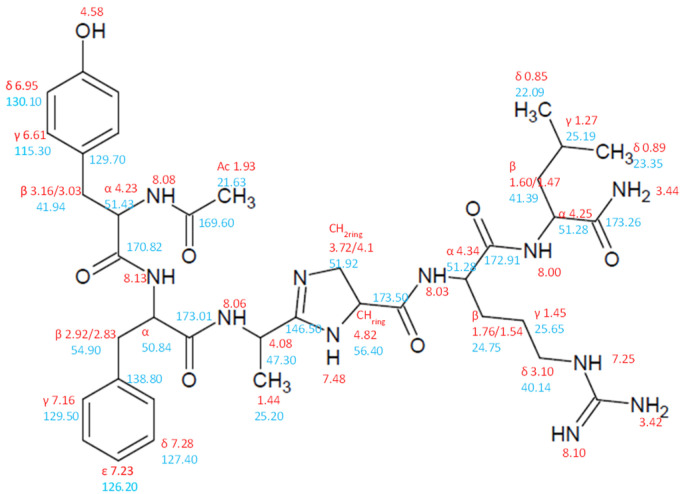
Structure of PPK-Jo with shifts of H (red) and C (blue) atoms displayed.

## Data Availability

The data presented in this study are available on request from the corresponding author.

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
