# Peer review of "Solid-Phase Synthesis of an Insect Pyrokinin Analog Incorporating an Imidazoline Ring as Isosteric Replacement of a trans Peptide Bond"

_molecules, 2021, doi:10.3390/molecules26113271_

Round 1

Reviewer 1 Report

The article entitled “Solid phase synthesis of an insect pyrokinin analog incorporating an imidazoline ring as isosteric replacement of a trans peptide bond” describes the synthesis of peptides containing an imidazoline function using a solid phase synthetic method. This synthetic methodology provides a useful alternative for the preparation of peptides with potential biological activity and thus, I recommend that the communication can be accepted for publication after minor revision.

In my opinion, the materials and methods section, in the form in which it is written, is a continuation of the results and discussion and should be included in that section.

The materials and methods are found in the supplementary material, so they can be transcribed to section 3. Alternatively, section 3 can only mention that the materials and methods are found in the supplementary material. The conclusions should also be in a section of their own.

The number of self-citations is high, 9 out of 16 references, therefore, if possible, some should be replaced by papers of other authors.

Reviewer 2 Report

This paper describes a solid-phase synthesis of PPK-Jo, a peptide mimic containing an imidazoline moiety. Overall, I find the SI quality is low and a few technical issues need to be resolved before this paper can be published.

1. The authors should provide an HPLC chromatogram to establish the purity of the final peptide.

2. Page 4 line 128, the author promised an HPLC chromatogram, but I could not find it.

3. Page S9 Figure 3. The NMR shows many peaks with fractional integration (such as 0.48, 0.26, 0.12, etc.). In addition, this spectrum seems to contain some isomers. Explanations must be provided.

4. Table 1 page S8 and the data in page S7 are from H and C spectra in DMSO. But these two spectra are missing.

Round 2

Reviewer 2 Report

I appreciate the authors' efforts and responses. While I still have my concerns about the fractional integrations in the 1H NMRs (Fig 3 as well as the new Fig 5 where the integrations seem to not match the numbers of hydrogen as tabulated on page S7), it is ultimately the authors' responsibilities to uphold the data quality.